# Structural basis of HapE[P88L]-linked antifungal triazole resistance in *Aspergillus fumigatus*

Peter Hortschansky[1] , Matthias Misslinger[2], Jasmin Mörl[2], Fabio Gsaller[2], Michael J Bromley[3], Axel A Brakhage[1], Michael Groll[4], Hubertus Haas[2], Eva M Huber[4]

Azoles are first-line therapeutics for human and plant fungal infections, but their broad use has promoted the development of resistances. Recently, a pan-azole–resistant clinical *Aspergillus fumigatus* isolate was identified to carry the mutation P88L in subunit HapE of the CCAAT-binding complex (CBC), a conserved eukaryotic transcription factor. Here, we define the mechanistic basis for resistance in this isolate by showing that the HapE[P88L] mutation interferes with the CBC's ability to bend and sense CCAAT motifs. This failure leads to transcriptional derepression of the *cyp51A* gene, which encodes the target of azoles, the 14-α sterol demethylase Cyp51A, and ultimately causes drug resistance. In addition, we demonstrate that the CBC-associated transcriptional regulator HapX assists *cyp51A* repression in low-iron environments and that this iron-dependent effect is lost in the HapE[P88L] mutant. Altogether, these results indicate that the mutation HapE[P88L] confers increased resistance to azoles compared with wt *A. fumigatus*, particularly in low-iron clinical niches such as the lung.

## Introduction

The global burden of aspergillosis exceeds 14 million people and mortality rates are especially high in patients with chronic and invasive diseases (Bongomin et al, 2017). The main class of therapeutics used to treat aspergillosis are azoles, in particular triazoles. Sub-optimal, widespread and long-term use of these drugs, however, has promoted the development of resistances. In some European centers, the levels of resistance exceed 20% and the U.S. Centers for Disease Control and Prevention have placed *Aspergillus fumigatus*, the primary etiological agent responsible for aspergillosis, on their watch list for antibiotic-resistant pathogens (https://www.cdc.gov/drugresistance/pdf/threats-report/2019-ar-threats-report-508.pdf). This worldwide development is of growing concern and demands a thorough understanding of the molecular mechanisms that contribute to drug resistance to support the development of alternative therapeutic strategies.

Recently, patient-acquired azole resistance of the human pathogenic mold *A. fumigatus* has been linked to the CCAAT-binding complex (CBC) (Arendrup et al, 2010; Camps et al, 2012a; Gsaller et al, 2016), a highly conserved and fundamental eukaryotic transcription factor (Bhattacharya et al, 2003). The core version of the CBC is a heterotrimer of the subunits HapB, HapC, and HapE that binds the CCAAT box, a promoter element present in about 30% of all eukaryotic genes (Bucher, 1990; Marino-Ramirez et al, 2004; Furukawa et al, 2020). The two histone-like subunits HapC and HapE associate with the DNA backbone and bend it in a nucleosome-like manner, whereas HapB with its sensor helix αS and its C-terminal anchor inserts into the minor groove and recognizes the CCAAT box (Huber et al, 2012; Nardini et al, 2013). Depending on the target gene and other transcriptional regulators, the CBC hereby either activates or inhibits gene expression.

In certain fungi such as *Aspergillus sp.*, a subset of genes are controlled by a more sophisticated variant of the CBC, termed CBC–HapX. This complex consists of HapB, HapC, and HapE, as well as two copies of HapX. HapX is a basic region leucine zipper (bZIP) that features an additional DNA-binding site 12 bps downstream of the CCAAT box (Hortschansky et al, 2017; Furukawa et al, 2020). CBC–HapX–controlled target genes are involved in iron homeostasis, storage, and consumption as well as ergosterol biosynthesis (Hortschansky et al, 2007; Gsaller et al, 2014, 2016). Ergosterol is a key component of fungal cell membranes and ensures their integrity as well as fluidity. Its biosynthesis involves the 14-α sterol demethylase Cyp51A, which is the primary target of azole-based antifungal drugs such as voriconazole (Odds et al, 2003; Becher & Wirsel, 2012; Monk et al, 2020).

Apart from mutations in the Cyp51A enzyme that prevent drug binding (Snelders et al, 2010), azole-resistant phenotypes can be based on efflux transporters (Fraczek et al, 2013) or on alterations of the *cyp51A* promoter (Snelders et al, 2011). In wild-type (wt) *A. fumigatus*, the *cyp51A* promoter contains binding sites for three

---

[1]Department of Molecular and Applied Microbiology, Leibniz Institute for Natural Product Research and Infection Biology (HKI), and Friedrich Schiller University Jena, Jena, Germany   [2]Institute of Molecular Biology/Biocenter, Innsbruck Medical University, Innsbruck, Austria   [3]Manchester Fungal Infection Group, Institute of Inflammation and Repair, University of Manchester, Manchester, UK   [4]Center for Integrated Protein Science Munich at the Department Chemistry, Technical University of Munich, Garching, Germany

Correspondence: eva.huber@tum.de

counteracting transcription factors: two inducers, the sterol regulatory element–binding protein SrbA (Gsaller et al, 2016) and the ATP-binding cassette transporter regulator AtrR (Paul et al, 2019), as well as a repressor, CBC–HapX (Gsaller et al, 2016). In azole-resistant *A. fumigatus*, however, duplication of a 34-mer region in the promoter (tandem repeat of 34 bps, TR34) creates additional binding sites for SrbA and AtrR, thereby leading to enhanced expression of the *cyp51A* gene, overproduction of the Cyp51A enzyme, and eventually to azole resistance (Gsaller et al, 2016; Paul et al, 2017).

Recently, in a patient infected with *A. fumigatus*, another mechanism of azole insensitivity has been discovered. The mutation leads to the amino acid change P88L in subunit HapE of the CBC, impairs the binding affinity of the complex to its target site, and prevents repression of the *cyp51A* gene (Camps et al, 2012a; Gsaller et al, 2016). This condition also leads to drug resistance by enhanced production of the Cyp51A enzyme, but how the mutant HapE[P88L] subunit alters

functioning of the CBC remained unknown. We here investigated the molecular mechanism of HapE[P88L]-mediated CBC dysfunction using in vivo and in vitro experiments. X-ray crystallographic analysis of the mutant CBC provided explanation for the reduced affinity of the CBC to its target DNA and significantly extended our current understanding of HapE[P88L]-induced azole resistance.

# Results

## In vivo analysis of *hapE*[P88L]-induced effects

Biological and physiological impacts of the HapE[P88L]-mutant CBC subunit were evaluated first in vivo by probing the ability of isogenic wt and *hapE*[P88L]-mutant isolates of *A. fumigatus* to grow under different

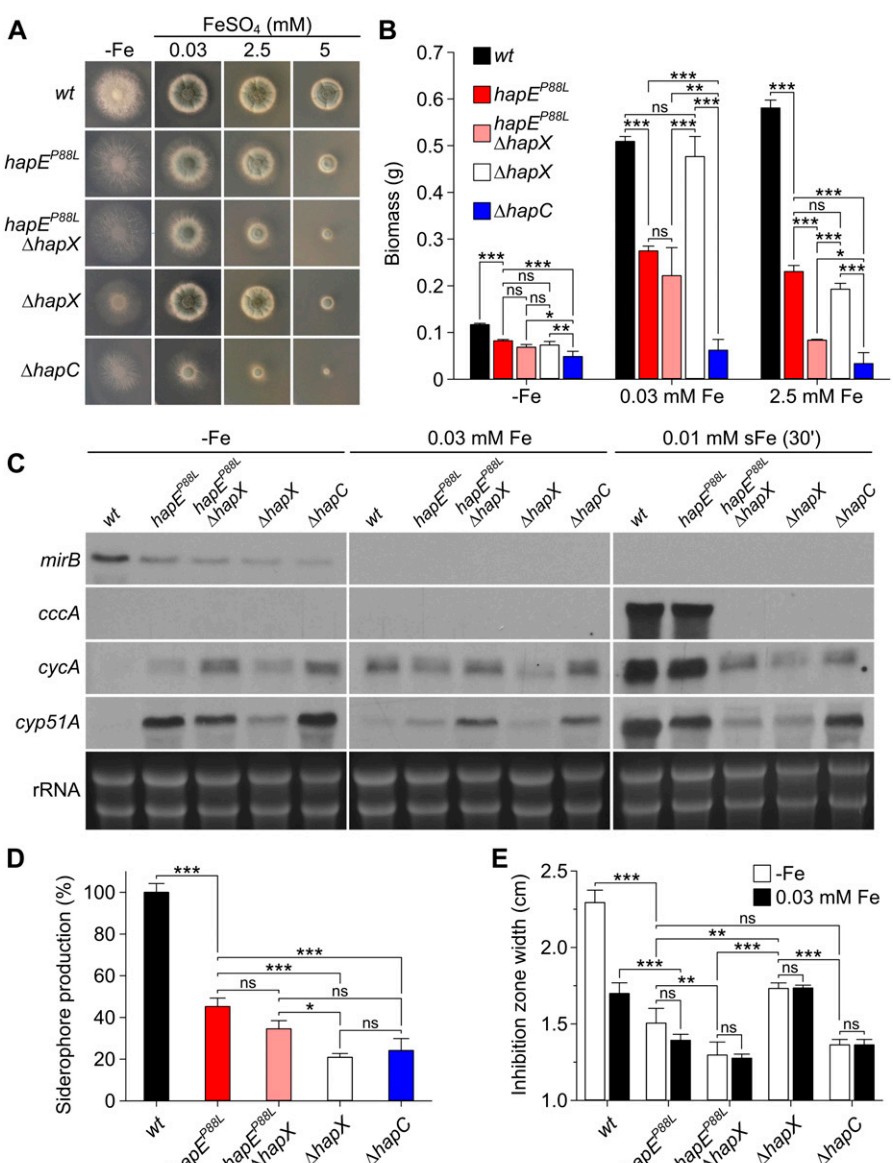

**Figure 1. Growth phenotyping, gene expression analysis, siderophore production, and azole resistance of *Aspergillus fumigatus* CBC mutants.**
**(A)** Growth pattern of *A. fumigatus* wild-type (*wt*), *hapE*[P88L], *hapE*[P88L]Δ*hapX*, Δ*hapX*, and Δ*hapC* strains on solid minimal medium containing different iron concentrations. Growth was evaluated after incubation at 37°C for 48 h. **(B)** Production of biomass in submersed cultures (liquid growth at 37°C for 24 h) during iron starvation (−Fe), iron sufficiency (0.03 mM $FeSO_4$, +Fe), and iron excess (2.5 mM $FeSO_4$). **(C)** Gene expression levels of the CBC and CBC–HapX target genes *mirB* (siderophore transporter), *cccA* (vacuolar iron transporter), *cycA* (cytochrome c), and *cyp51A* (14-α sterol demethylase Cyp51A) under the indicated iron conditions. Northern blot analyses were performed from liquid cultures grown at 37°C for 20 h under iron starvation (−Fe) or iron sufficiency (0.03 mM $FeSO_4$). Alternatively, mycelia were shifted for 30 min from −Fe to iron sufficiency (0.01 mM $FeSO_4$, sFe) to generate short-term iron excess. As a loading control, ribosomal (r)RNA levels are shown below. **(D)** Siderophore production (triacetylfusarinine C and fusarinine C) in mutant *A. fumigatus* strains compared with wt in the absence of iron. **(E)** Iron-dependent azole resistance of *A. fumigatus* mutants. Voriconazole (10 µl of 320 µg/ml) was spotted on agar plates inoculated with *A. fumigatus* spores, and the width of the inhibition zone was observed as a measure of drug resistance after 48 h. The narrower the inhibition zone was, the more resistant the strains were. Data information: In (B, D, E), data are presented as the mean and SD of three biological replicates and analyzed by one-way ANOVA with Tukey's multiple comparison test (*$P \leq 0.05$; **$P \leq 0.01$; ***$P \leq 0.001$; ns, not statistically significant). Source data are available for this figure.

conditions. As the CBC–HapX complex controls fungal adaption to varying iron concentrations (Schrettl et al, 2010; Gsaller et al, 2014), iron depletion, sufficiency, and excess were tested. In all settings, growth of the mutant was clearly impaired compared with wt *A. fumigatus*. Deletion of the gene encoding the HapX subunit in the *hapE^P88L* background further aggravated the phenotype when compared with the respective single mutants (Fig 1A). The phenotype of the *hapE^P88L* mutant was less severe than for an isogenic strain lacking a functional CBC (Δ*hapC*). Similar results were observed when monitoring the fungal biomass obtained from liquid cultures under various iron concentrations (Fig 1B). In summary, the *hapE^P88L* mutation severely impairs viability parameters of *A. fumigatus* and, in particular, the tolerance to low and high iron stress.

To sequester iron from the surroundings, *A. fumigatus* secretes chelators termed siderophores (Haas, 2014), whose re-uptake is mediated by siderophore transporters such as MirB. During iron starvation (–Fe), the wt CBC–HapX complex stimulates biogenesis of MirB to promote acquisition of the metal. In the *hapE^P88L* mutant, however, the strength of gene induction by the CBC was considerably lower (Fig 1C). Furthermore, the *hapE^P88L* strain produced at least 50% less of the extracellular siderophores triacetylfusarinine C and fusarinine C than wt (Fig 1D). In low-iron environments, wt *A. fumigatus* also down-regulates *cycA* and *cyp51A* genes, which encode the iron-dependent proteins cytochrome c and 14-α sterol demethylase Cyp51A, respectively, to restrict nonessential iron use (Fig 1C). Both genes, however, showed significant expression in the *hapE^P88L* mutant, indicating a transcriptional deregulation. Strikingly under short-term iron sufficiency (sFe), effects of the *hapE^P88L* mutation were less discernible. Transcription of the *cccA* gene, coding for the vacuolar iron importer, was not affected in the *hapE^P88L* strain but completely abrogated in Δ*hapX* backgrounds (Fig 1C). A similar,

although weaker, tendency was found for *cycA* and *cyp51A*. As the promoters of all three of these genes are strong targets of the wt CBC–HapX complex (Furukawa et al, 2020; Gsaller et al, 2014), HapX might compensate the deleterious effects of the mutant HapE subunit and enable their transcription. In agreement, inactivation of HapX in the *hapE^P88L* mutant decreases viability (Fig 1A). Together, these results suggest that HapX plays a dominant role in stabilizing the DNA–regulator complex and that the mutation *hapE^P88L* affects transcription of a subset of CBC targets within the genome including *cyp51A*. Considering that the CBC controls expression of about 30% of all eukaryotic genes (Bucher, 1990; Furukawa et al, 2020), the mutant HapE subunit might cause dysregulation of many biochemical pathways and provoke the observed severe growth retardation of *A. fumigatus*.

Next, we tested the resistance of *A. fumigatus* to the broad-spectrum antifungal medication voriconazole. Consistent with iron-controlled expression of the *cyp51A* gene (Fig 1C), resistance of wt *A. fumigatus* to voriconazole was iron dependent. Furthermore, loss of CBC function (Δ*hapC* or *hapE^P88L*) or HapX abrogated this effect. During iron starvation, *hapE^P88L* strains were considerably more resistant to the drug than wt, revealing that the transcriptional derepression observed for *cyp51A* in the mutant *A. fumigatus* isolate correlates with increased resistance. In agreement, in the presence of iron, which stimulates *cyp51A* expression, the effect was less pronounced (Fig 1E).

## In vitro studies with HapE^P88L-mutant CBC

To evaluate whether the mutant CBC is able to bind target DNAs in vitro, we performed surface plasmon resonance (SPR) experiments with purified wt and mutant CBCs, as well as various 25-bp long nucleic acid duplexes. Consistent with results from the *cyp51A* promoter (Gsaller

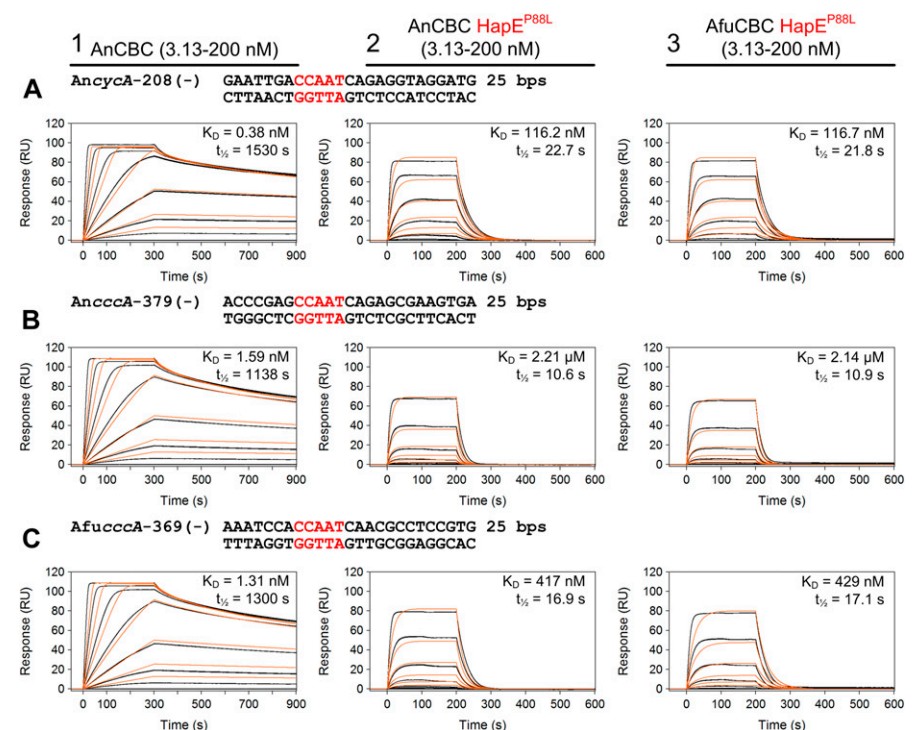

**Figure 2. DNA affinity of wt and mutant CBC preparations.**

**(A, B, C)** Surface plasmon resonance-binding analysis of wt (panel 1) and HapE^P88L-mutant *Aspergillus nidulans* (An) (panel 2) as well as *Aspergillus fumigatus* (Afu) CBCs (panel 3) (3.13–200 nM) to different DNA duplexes. Sequences were derived from either the *A. nidulans cycA* (A) or *cccA* promoter (B) or the *A. fumigatus cccA* regulatory element (C). Binding responses of the indicated CBC concentrations injected in duplicate (black lines) are overlaid with the best fit derived from a 1:1 interaction model, including a mass transport term (red lines). $K_D$ constants and half-lives of the complexes are provided for each panel. Affinities of the mutant CBC are decreased by a factor of about 140–1,300 depending on the target sequence.

et al, 2016), HapE[P88L]-mutant CBCs showed a drastic decrease in affinity for *cycA* ($K_D$ increases by factor 140) and *cccA* (up to factor 1,390) CCAAT sequences (Fig 2). Similarly, the half-lives of the protein–DNA assemblies were strongly reduced (factor 67 for *cycA* and at least factor 76 for *cccA*). This result confirms that in the HapE[P88L] context, transcriptional control by the CBC is defective. Furthermore, this effect was the same for *A. fumigatus* (Afu) and *Aspergillus nidulans* (An) CBC (Fig 2), indicating that the mechanism of HapE[P88L] transcriptional deregulation is the same across species. Regarding the wt-like expression profile of the *cccA* gene in the *hapE[P88L]* setting during a short-term shift from iron starvation to iron sufficiency (sFe; Fig 1C), we additionally investigated the effect of HapX on the DNA-binding affinity of mutant CBC. SPR coinjection assays revealed that the half-life of the ternary CBC[P88L]–*cccA*–HapX complex is increased sevenfold compared with the binary CBC[P88L]–*cccA* particle (Fig 3). Hence, it appears that HapX is at least partially able to restore the ability of the CBC to bind to its recognition site.

### Structural examination of HapE[P88L]-mutant CBC

To investigate how the P88L mutation in HapE alters the DNA-binding capacity of the CBC and ultimately confers drug resistance, we attempted the crystallization of wt and mutant CBC from *A. fumigatus* (Fig S1A). Structures of the Afu_CBC could be solved in complex with double-stranded 23-bp-long DNA fragments derived from the promoter sequences of either *cycA* (2.6 Å resolution, Table 1 and Fig S1B) or *cccA* (2.3 Å resolution, Table 1 and Fig S1B). In addition to our previously determined An_CBC–*cycA* crystal structure (Huber et al, 2012), we also obtained data on the An_CBC in

complex with the *cccA* DNA fragment (2.2 Å resolution, Table 1 and Fig S1B). Superpositions of the Afu_CBC–*cycA* and the Afu_CBC–*cccA* complexes as well as the corresponding *A. nidulans* proteins indicated high structural similarity, suggesting that complex arrangement and DNA bending are uniform among these species and independent of the nucleic acid sequence and the target gene promoter site (root-mean-square deviation [rmsd] 0.153 Å for *A. fumigatus*, 0.246 Å for *A. nidulans*, 0.306 Å for *cycA*, and 0.369 Å for *cccA* complexes; Fig S2A).

Azole-resistant Afu_CBC[P88L] protein preparations, however, did not stably associate with Afu_*cccA* promoter-derived double-stranded DNA, as confirmed by size exclusion chromatography (SEC) (Fig 4A). The residual complex affinity of 429 nM (Fig 2C) probably was not high enough to counteract the shearing forces during chromatography. In addition, Afu_CBC[P88L] failed to crystallize in the presence of An_*cycA* promoter DNA (Fig 4B). We, therefore, switched organisms and created the HapE[P88L]-mutant *A. nidulans* CBC. Despite reduced affinity (116 versus 0.83 nM for wt; Fig 2A), we obtained a SEC-stable complex for this variant with the 23-bp-long *cycA* promoter fragment (Fig 4C) and elucidated its X-ray structure at 2.3 Å resolution (Table 1). With the previously solved wt An_CBC–*cycA* complex in hands (PDB ID 4G92), direct comparison with the mutant protein–DNA structure was possible. In contrast to the wt CBC that binds the DNA fragment in a 1:1 stochiometry (Huber et al, 2012), three CBC[P88L] complexes associate with one DNA double strand encoding a single CCAAT-binding motif (Fig 5A). Notably, the orientations of the three CBCs relative to the CCAAT-binding motif deviate from each other, and all differ from the wt structure (Figs 5A and S3).

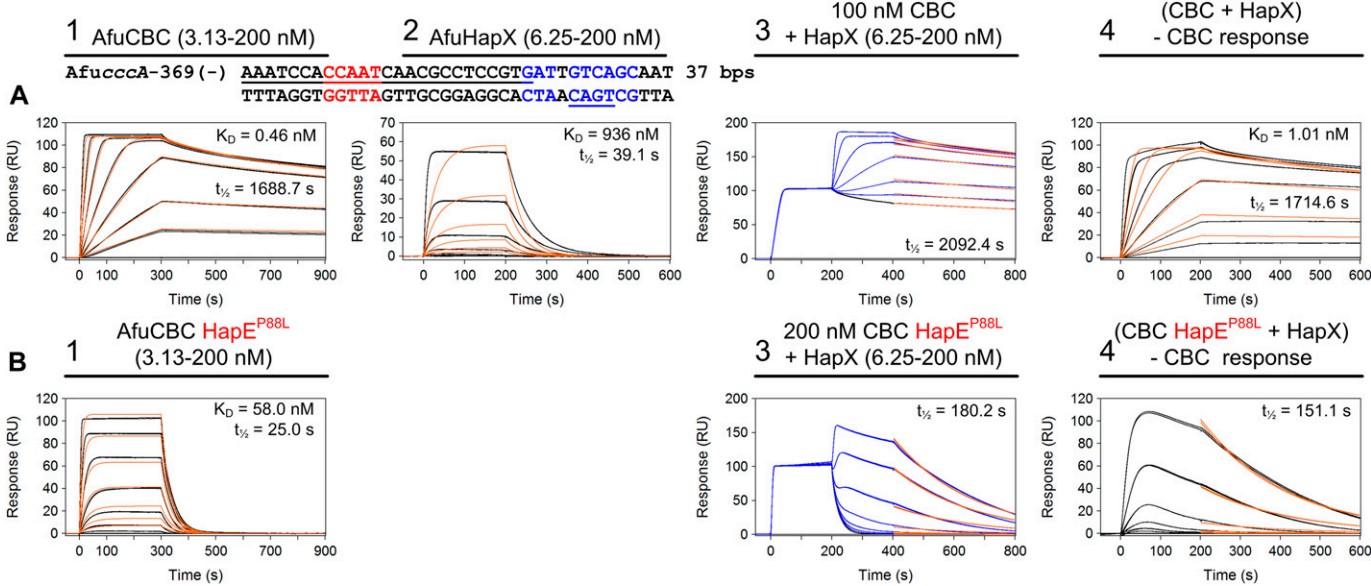

**Figure 3. Cooperative CBC–HapX binding stabilizes DNA interaction of the HapE[P88L]-mutant CBC.**
**(A, B)** Surface plasmon resonance analyses are shown for binding of wt CBC (A) or HapE[P88L]-mutant CBC (B) to DNA (panel 1), HapX to DNA (panel 2), and HapX to preformed CBC–DNA complexes (panel 3). The sequence of the immobilized DNA duplex is derived from the *Aspergillus fumigatus cccA* promoter. Nucleotides (nts) underlined in black are covered by the CBC (Huber et al, 2012), and nts marked in blue represent the HapX consensus binding site (Gsaller et al, 2014). Binding responses of the indicated CBC or HapX concentrations injected in duplicate (black lines) are overlaid with the best fit derived from a 1:1 interaction model, including a mass transport term (red lines). Binding responses of CBC–DNA–HapX ternary complex formation (panel 3, blue lines) were obtained by concentration-dependent co-injection of HapX on preformed binary CBC–DNA complexes after 200 s within the steady-state phase. Sensorgrams in panel 4 depict the association/dissociation responses of HapX on preformed CBC–DNA and were generated by CBC response subtraction (co-injection of buffer) from HapX co-injection responses. Dissociation constants ($K_D$) and half-lives of the complexes are plotted inside the graphs.

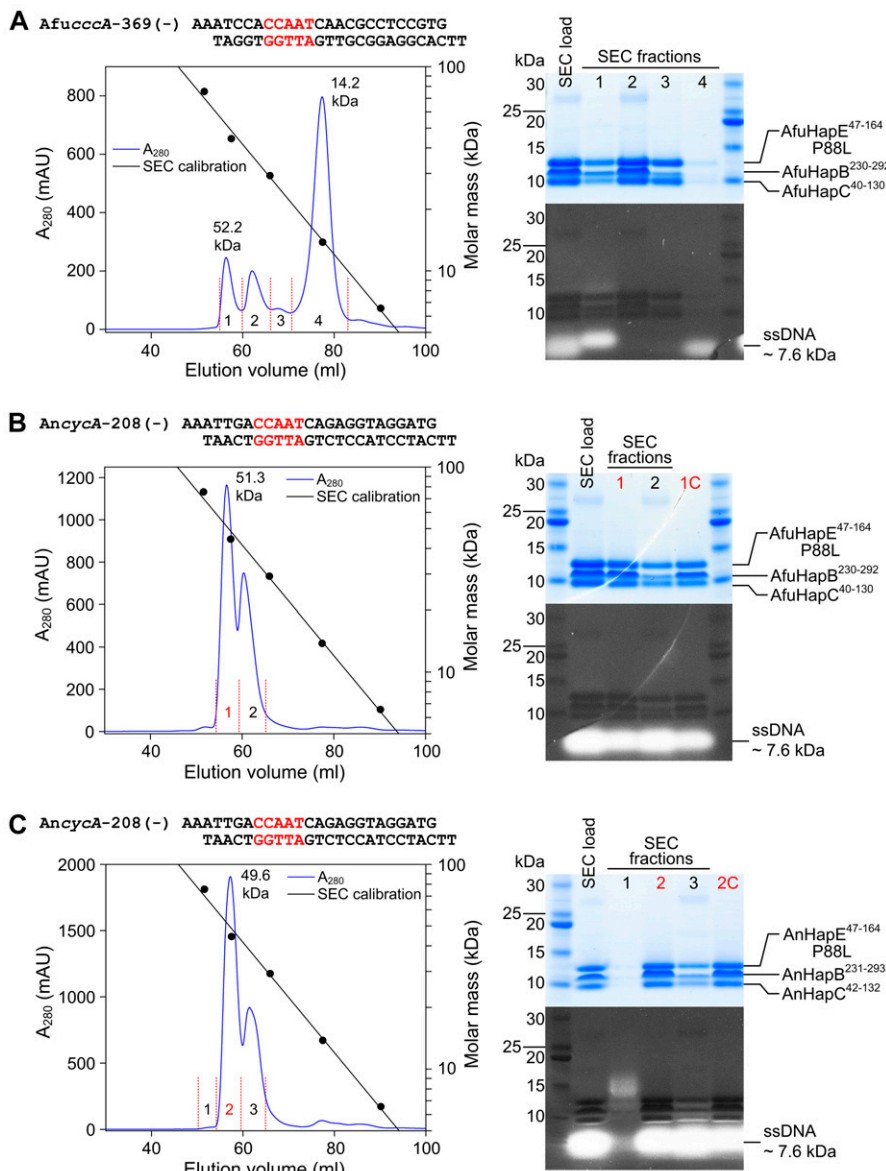

**Figure 4. Reconstitution of CBC–DNA complexes for crystallization.**
**(A, B, C)** Size exclusion profiles and SDS–PAGE analysis of HapE[P88L]-mutant CBC–DNA complexes from (A, B) *Aspergillus fumigatus* and (C) *Aspergillus nidulans*. The strongest CBC target sites from *A. fumigatus cccA* (A) and *A. nidulans cycA* (B, C) promoter sequences were chosen as DNA duplexes. Size exclusion chromatography (SEC) fractions that were subjected to crystallization after a subsequent concentration step are marked in red (samples labeled 1C and 2C in the respective SDS–PAGE gels). SDS–PAGE gels were stained for protein with the GelCode Blue Stain Reagent (upper panels) before DNA staining with SYBR Gold Nucleic Acid Gel Stain (lower panels).
Source data are available for this figure.

Intriguingly, the mutant CBCs do not significantly bend the DNA as it was previously noted for the wt protein–DNA complex (Huber et al, 2012) (Fig 5B). Whereas the wt CBC induces a bending angle of 68° (Huber et al, 2012), the respective parameter for the mutant CBC–DNA complex is 9.3°. Reduced DNA curvature results from an altered binding mode of the CBC[P88L] complexes to the DNA–sugar–phosphate backbone. Actually, for all three CBC[P88L] proteins in the asymmetric unit, different interaction patterns with the DNA were observed and most of them are based on hydrogen bonds between protein main chain amides and phosphate moieties of the DNA (Fig S3).

Structural superposition proved that the three copies of the protein complex are identical and comparison with the wt CBC coordinates (PDB ID 4G91 [Huber et al, 2012]) illustrated that the mutation P88L neither disrupts the subunit fold nor complex assembly (rmsd < 0.169 Å). However, in each of the mutant CBCs, only

the αN helix of HapB was defined in the $2F_O$-$F_C$ electron density map, whereas the αS sensor helix, which usually inserts into the DNA double strand and thereby confers sequence specificity to the CBC (Huber et al, 2012), was disordered (Figs 5A and B and 6).

The site of mutation, Pro88, forms the boundary between loop L0 and helix α1 of subunit HapE, and the succeeding residues 89–94 undergo hydrogen bond interactions with the sugar–phosphate backbone of the nucleic acid in the wt CBC–*cycA* crystal structure (Huber et al, 2012) (Fig 5B and C). Substitution of Pro88 by Leu leads to N-terminal elongation of helix α1 by approximately a half turn (Figs 5D and 6). Superposition of the wt CBC–*cycA* complex with the mutant CBC[P88L] protein indicates that the mutation-induced extension of helix α1 clashes with the bent conformation of the DNA observed for the wt CBC (Fig S4). Thus, it appears that the mutation P88L prevents histone-like DNA binding by the CBC and insertion of HapB's αS helix into the

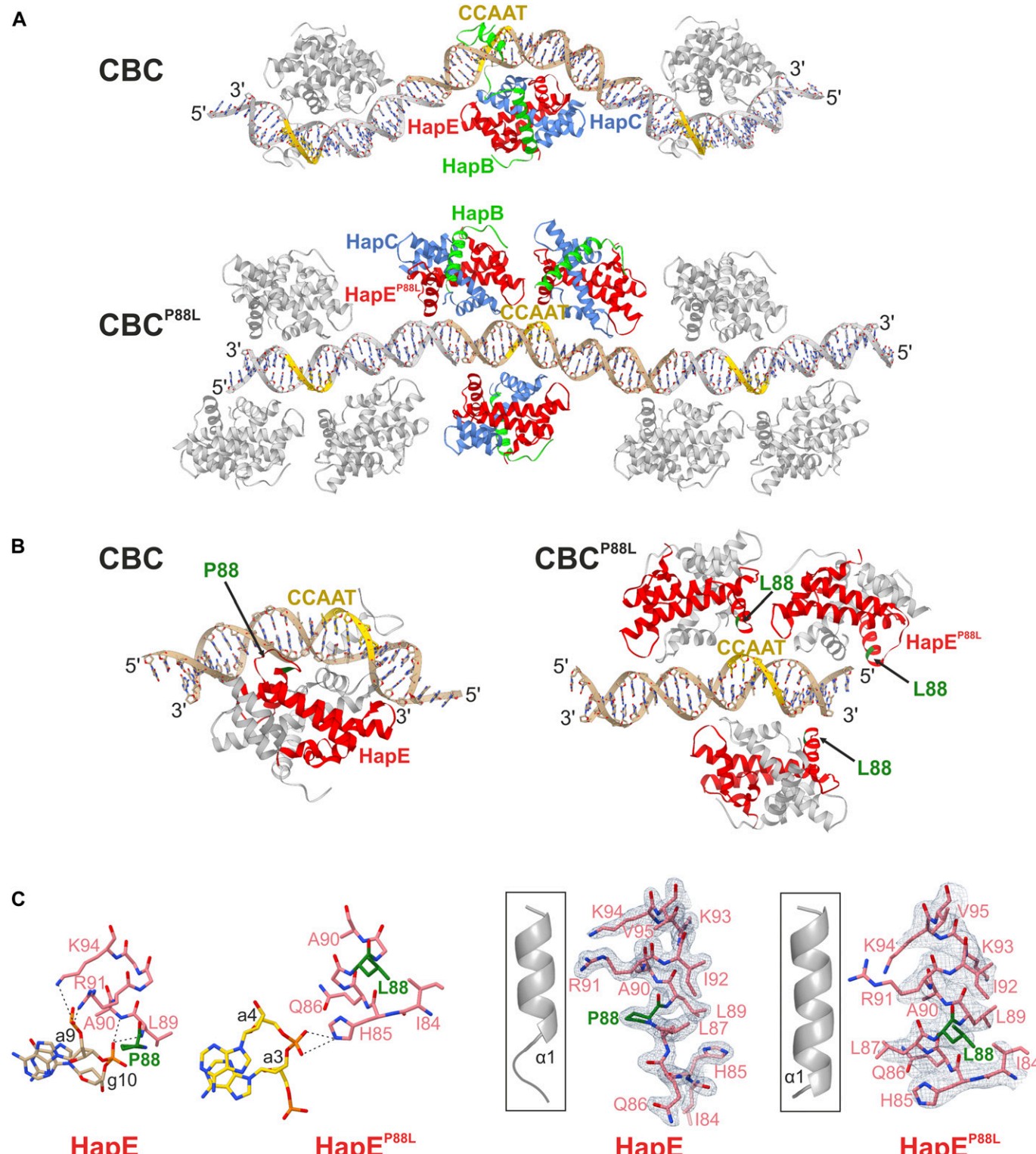

**Figure 5. Structural comparison of wt and HapE^P88L mutant An_CBC–cycA complexes.**
**(A)** Ribbon illustration of CBC–DNA complexes. The 23-bp *cycA* DNA fragments and the protein assemblies forming the asymmetric units of the crystals are color-coded, whereas crystallographic symmetry mates are shown in gray. Because of the 5′ AA-TT overhangs, the DNA duplexes arrange as fibers in the crystal lattice. The CCAAT-binding motif is colored yellow, whereas HapB, HapC, and HapE subunits are depicted in green, blue, and red, respectively. The mutant CBC does not induce significant DNA bending and fails to interact sequence-specifically with the DNA. Because of nonspecific DNA-binding events, the stoichiometry of mutant CBC complexes to DNA fragments is increased to 3:1 compared with 1:1 for wt CBC. **(B)** In wt An_CBC–*cycA*, Pro88 (green; black arrow) of HapE (red) is located adjacent to the DNA. The P88L mutation disrupts the interaction with the DNA and its bending. Structures are rotated by 180° along the y-axis compared with panel (A). **(C)** Hydrogen bond interactions

CCAAT-binding motif. Altogether, the affinity of mutant CBC complexes for CCAAT-binding sites is severely reduced and this loss of sequence specificity leads to the association of three transcription factors per DNA double strand (Fig 5A). The residual ability to bind DNA, however, is insufficient to properly position the CBC in the promoter region of the *cyp51A* gene and to repress its transcription. The uncontrolled expression of *cyp51A* leads to overproduction of the 14-α sterol demethylase Cyp51A and eventually renders commonly applied azole concentrations inactive.

## Discussion

Infections with azole-resistant *A. fumigatus* are of growing concern in clinics. Azoles are the only orally available antifungals (Lelievre et al, 2013), and alternative agents to treat invasive aspergillosis are scarce. Hence, patients suffering from drug-resistant invasive aspergillosis face mortality rates of up to 100% (Meis et al, 2016). The widespread use of triazole-based fungicides in agriculture and export of crops are likely to have contributed to the emergence and spread of resistance (Snelders et al, 2008; Verweij et al, 2009; Camps et al, 2012b; Chowdhary et al, 2013; Bowyer & Denning, 2014; Dunne et al, 2017). This is supported by the fact that patients are frequently diagnosed with azole-resistant invasive aspergillosis despite they have not received antifungal treatment before.

The most common mechanisms of resistance detected in invasive aspergillosis include mutations in the gene which encodes the target of azole compounds, the enzyme Cyp51A (Snelders et al, 2010), or duplications of the promoter region that regulates *cyp51A* expression (Snelders et al, 2011). Here, we investigated how the recently discovered mutation P88L in subunit HapE of the CBC, a ubiquitous transcription factor, confers azole resistance to *A. fumigatus* (Camps et al, 2012a).

We show that azole resistance and iron homeostasis are inextricably linked through the action of the CBC and its accessory subunit HapX. In particular, in wt *A. fumigatus*, azole resistance is decreased under low-iron conditions which is consistent with reduced *cyp51A* expression. This low-iron–mediated azole sensitivity is abolished in HapX-deficient backgrounds and in the *hapE^P88L* mutant. We also found that a *hapE^P88L* mutant of *A. fumigatus* is less resistant to iron starvation as well as iron overload because of altered gene regulation by the CBC. In vitro SPR analysis revealed that HapE^P88L mutant CBCs poorly bind to CCAAT boxes in general. These results agree with the reduced CBC-affinity reported for the *cyp51A* gene (Gsaller et al, 2016) and mutagenesis experiments on the human HapE homolog NF-YC, which showed residues 43–45 (corresponding to 87–89 in *A. fumigatus* and *A. nidulans*) to be essential for DNA binding (Zemzoumi et al, 1999). For this reason, attempts to crystallize the HapE^P88L–mutant CBC from *A. fumigatus* in complex with DNA failed. Likewise, in the absence of nucleic acid, Afu_CBC did not crystallize. The primary sequences of Afu_CBC and An_CBC only differ by two Val to Ile replacements in subunit HapE (Fig S1A). As visualized by the wt Afu_CBC–*cccA* and wt Afu_CBC–*cycA*

complex structures, these conserved amino acid variations cause a slight shift of the N-terminal αN helix of HapE that might enhance mobility and prevent crystallization in the absence of a high-affinity DNA ligand (Fig S2B). We, therefore, focused on the *A. nidulans* CBC. Structures of wt and HapE^P88L-mutant An_CBCs in complex with DNA visualized that reduced curvature of the nucleic acid is the primary cause for the low affinity to the HapE^P88L mutant CBC. The mutation P88L elongates helix α1 of subunit HapE and sterically interferes with DNA bending (Fig S4). This observation agrees with the reported propensity of proline to often N-terminally cap α-helices (Richardson & Richardson, 1988; Kim & Kang, 1999; Cochran et al, 2001) and its helix breaker function in soluble proteins (Chou & Fasman, 1974) as well as the tendency of leucine to be part of α-helices (Fujiwara et al, 2012). Because of the altered bending angle of the DNA, the sequence-specific HapB subunit fails to find the CCAAT motif, resulting in its structural disorder and the random positioning of CBC complexes on the DNA via electrostatic interactions with the sugar–phosphate backbone. The crucial importance of Pro88 for DNA curvature and high-affinity binding is underpinned by its strict conservation and the X-ray structure of the human CBC homolog, the NF–Y complex (Nardini et al, 2013).

Despite these structural changes in the HapE^P88L mutant, in vivo and SPR experiments suggest that in the presence of HapX, the affinity on CBC–HapX target sequences is partially retained. Although the basic region leucine zipper HapX can act as a transcription factor only when bound to the CBC, it has an own DNA recognition motif downstream of the CCAAT box (Gsaller et al, 2014; Hortschansky et al, 2015; Furukawa et al, 2020). It is, therefore, conceivable that in the *hapE^P88L* mutant strain, HapX guides CBC–HapX complexes to the nucleic acid and by binding to its target sequence may enable correct positioning of the CBC near the CCAAT sequence. This way, the HapB subunit may be able to recognize the CCAAT motif and to insert into the DNA double strand (Huber et al, 2012; Nardini et al, 2013). In agreement, expression of the strong CBC–HapX target *cccA* is not affected by the *hapE^P88L* mutation, and inactivation of HapX in the *hapE^P88L* background further reduced the growth ability. We, therefore, suppose that promoters encoding solely the CCAAT box are more severely affected by the mutant HapE subunit and are more likely to lose their transcriptional control than those featuring in addition a HapX-binding site. However, depending on the promoter sequence, other interaction partners of the CBC might influence the strength of DNA binding and hence the level of transcription as well.

In contrast to other azole resistance mechanisms, inactivation of the CBC, either by deletion of *hapC* (Gsaller et al, 2016) or by the point mutation P88L in HapE, significantly attenuates virulence of *A. fumigatus* (Arendrup et al, 2010). The reduced viability and pathogenicity of *hapE^P88L*-mutant *A. fumigatus* may be the reason why this mutation has so far only been identified in a clinical isolate and not in the environment. Nonetheless, because in every second patient, with azole resistance, the molecular mechanism is not mediated by Cyp51A and of unknown origin (Bueid et al, 2010), CBC-linked drug

of α1 helices of wt and mutant HapE with adjacent DNA backbones (dotted lines) differ because of the mutation P88L (green). Similarly, the orientation of the DNA as well as the site of interaction with the protein are distinct. Protein–DNA contacts are shown for one mutant CBC complex. Interactions of the remaining two CBC^P88L complexes with the DNA are provided in Fig S3. **(D)** 2F_O-F_C electron density maps are shown as gray meshes (contoured to 1σ) for the amino acid residues 84–95 of wt and mutant HapE. Leu88 (green) elongates the N-terminal part of helix α1 by a half turn compared with Pro88 (green).

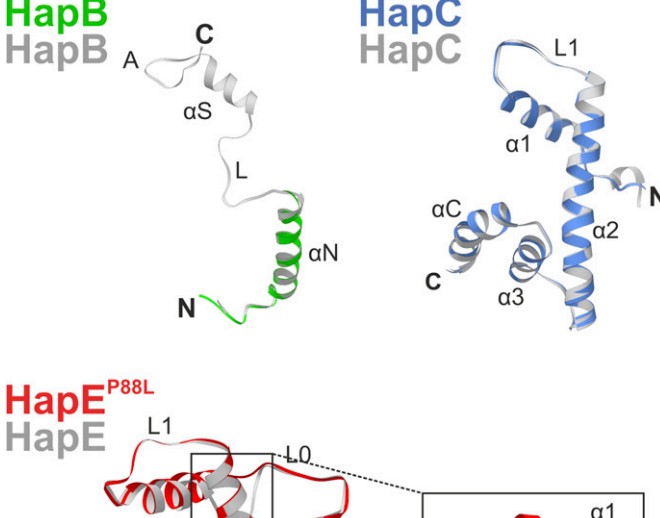

**Figure 6. Superposition of Hap subunits forming wt and HapE^P88L-mutant An_CBC–cycA complexes.**
Structural superposition of Hap subunits of the wt CBC (gray; PDB ID 4G92) and HapE^P88L-mutant CBCs (colored). In the mutant complex, the C-terminal αS helix and the anchor motif of subunit HapB (green) are disordered because of missing interactions with the DNA double strand (Fig 5A). Structural changes induced by P88L (green; arrows) in subunit HapE (red) are depicted.

resistance may also still be underexplored. Although other human pathogenic fungi such as *Aspergillus flavus* and *Aspergillus terreus* have not been reported to be azole resistant, it is alarming that a patient infected with the rather avirulent *hapE^P88L* mutant *A. fumigatus* strain died because of treatment failure (Camps et al, 2012a). The continuous rise in patients not responding to azoles and the identification of novel resistance mechanisms urgently demand for the development of novel agents for crop protection and clinical applications. Our data indicate that although agents targeting iron homeostasis by interfering with CBC–HapX function could be of significant value, they may be antagonistic with existing azole antifungals.

# Materials and Methods

### Generation of mutant *A. fumigatus* strains

Strains A1160P+ (wt), *hapE^P88L*, and Δ*hapC* have been described previously (Gsaller et al, 2016). To inactivate *hapX*, a construct containing a pyrithiamine resistance cassette was amplified from genomic DNA of a *hapX*-deficient strain (strain Δ*hapX*, background AfS77 [Gsaller et al, 2014]) with primers oAfhapX-1 (AGC GAC TAT AGC CGG ATG) and oAfhapX-2 (CCT TGG GTC TTG AAG CTT GCG) and transformed into an A1160P+ wt or

*hapE^P88L* recipient strain, respectively. Homologous recombination events yielded strains Δ*hapX* and *hapE^P88L*Δ*hapX*.

### Growth analysis of *A. fumigatus*

Growth assays were performed in Aspergillus minimal medium (1% [wt/vol] glucose, 20 mM glutamine, salt solution, and iron-free trace elements) according to previous reports (Pontecorvo et al, 1953).

### Measurement of siderophore production

Fungal strains were grown in liquid cultures under iron limitation conditions. After 24 h, the culture supernatants were transferred to new reaction tubes and saturated with FeSO$_4$. Next, 0.2 volumes of phenol:chloroform:isoamyl alcohol (25:24:1, PCI) was added for extraction of total extracellular siderophores (TAFC and FsC). After centrifugation, the PCI phase was mixed with five volumes of diethylether and one volume of water. In the last step, the upper diethylether containing phase was removed, and the amount of TAFC + FsC in the aqueous phase was quantified spectrophotometrically (NanoDrop2000; Thermo Fisher Scientific) using a molar extinction coefficient of $\varepsilon$ = 2,996 M$^{-1}$ cm$^{-1}$ at 440 nm.

### Overproduction and purification of recombinant CBC complexes

The *A. nidulans* CBC consisting of HapB$^{231–293}$, HapC$^{42–132}$, and HapE$^{47–164}$ was produced and purified as described (Gsaller et al, 2014). Briefly, synthetic genes coding for the conserved core domains of HapB, HapC, HapE, or HapE^P88L were sequentially cloned in the pnCS vector for expression of a polycistronic transcript (Diebold et al, 2011). The expression plasmids were transformed in *Escherichia. coli* BL21(DE3). After overnight autoinduction and cell lysis, the heterotrimeric wt CBC and the CBC^P88L mutant were purified to homogeneity by subsequent cobalt chelate affinity and SEC. The equivalent *A. fumigatus* wt and HapE^P88L mutant CBCs (HapB$^{230–292}$, HapC$^{40–130}$, and HapE$^{47–164}$) were produced the same way. Size exclusion fractions containing pure CBCs were pooled based on SDS–PAGE analysis, concentrated by ultrafiltration (Amicon Ultra-15 10K centrifugal filter device; Millipore) to 16–18 mg ml$^{-1}$, aliquoted, flash-frozen in liquid nitrogen, and stored at −80°C.

### SPR measurements

Real-time SPR protein–DNA interaction measurements were performed by previously published protocols (Gsaller et al, 2016). Notably, for cooperative CBC–HapX binding analysis measured by SPR co-injection on the *A. fumigatus cccA* promoter motif, *A. fumigatus* wt and HapE^P88L-mutant CBCs consisting of the HapB$^{230–299}$, HapC$^{40–137}$, and HapE$^{47–164}$ subunits were used. The *A. fumigatus* HapX$^{24–158}$ bZIP peptide (covering the CBC-binding domain, basic region, and coiled-coil domain) was produced and purified as previously described (Gsaller et al, 2014).

### Preparation of CBC–DNA complexes for crystallization

Oligonucleotides were produced by chemical synthesis of the forward and reverse strands (Biomers). These oligonucleotides were dissolved in annealing buffer (10 mM Tris/HCl and 50 mM NaCl, pH 8.0) at a

**Table 1. X-ray data collection and refinement statistics.**

| | Afu_CBC–*cycA* | Afu_CBC–*cccA* | An_CBC–*cccA* | An_CBC HapE[P88L]–*cycA* |
|---|---|---|---|---|
| Crystal parameter | | | | |
| Space group | P2₁2₁2₁ | P2₁2₁2₁ | P2₁2₁2₁ | P2₁2₁2₁ |
| Cell constants | a = 51.57 Å | a = 51.48 Å | a = 65.68 Å | a = 72.15 Å |
| | b = 75.51 Å | b = 83.74 Å | b = 72.01 Å | b = 103.50 Å |
| | c = 142.94 Å | c = 148.42 Å | c = 148.45 Å | c = 159.71 Å |
| Subunits/AU[a] | 1 Af_CBC | 1 Af_CBC | 1 An_CBC | 3 An_CBCs |
| | 1 DNA duplex | 1 DNA duplex | 1 DNA duplex | 1 DNA duplex |
| Data collection | | | | |
| Beam line | X06SA, SLS | X06SA, SLS | X06SA, SLS | X06SA, SLS |
| Wavelength (Å) | 1.0 | 1.0 | 1.0 | 1.0 |
| Resolution range (Å)[b] | 48–2.6 (2.7–2.6) | 48–2.3 (2.4–2.3) | 48–2.2 (2.3–2.2) | 49–2.3 (2.4–2.3) |
| No. of observations | 63,420 | 157,892 | 169,021 | 271,549 |
| No. of unique reflections[c] | 17,526 | 29,239 | 36,047 | 52,851 |
| Completeness (%)[b] | 98.4 (99.6) | 99.7 (99.8) | 98.7 (99.3) | 98.0 (98.4) |
| $R_{merge}$ (%)[b,d] | 4.5 (50.9) | 4.4 (58.9) | 5.1 (56.5) | 7.0 (59.8) |
| $I/\sigma$ (I)[b] | 19.0 (2.6) | 21.2 (2.9) | 16.0 (2.2) | 10.5 (1.7) |
| Refinement (REFMAC5) | | | | |
| Resolution range (Å) | 30–2.6 | 30–2.3 | 30–2.2 | 30–2.3 |
| No. of refl. working set | 16,636 | 27,765 | 34,232 | 50,185 |
| No. of refl. test set | 876 | 1,461 | 1,802 | 2,641 |
| No. of non-hydrogen | 3,132 | 3,167 | 3,357 | 6,811 |
| Solvent ($H_2O$, $Cl^-$) | 25 | 69 | 124 | 61 |
| $R_{work}/R_{free}$ (%)[e] | 19.2/23.2 | 18.7/21.1 | 19.5/21.8 | 22.1/25.4 |
| rmsd bond/angle (Å)/(°)[f] | 0.002/0.991 | 0.003/1.011 | 0.002/1.033 | 0.007/1.331 |
| Average B-factor (Å²) | 64.0 | 64.2 | 55.5 | 68.2 |
| Ramachandran plot (%)[g] | 98.0/2.0/0.0 | 98.3/1.7/0.0 | 98.5/1.5/0.0 | 97.5/2.5/0.0 |
| PDB accession code | 6Y35 | 6Y36 | 6Y37 | 6Y39 |

[a]Asymmetric unit.
[b]The values in parentheses for resolution range, completeness, $R_{merge}$, and $I/\sigma$ (I) correspond to the highest resolution shell.
[c]Data reduction was carried out with XDS and from a single crystal. Friedel pairs were treated as identical reflections.
[d]$R_{merge}(I) = \Sigma_{hkl}\Sigma_j|I(hkl)_j - <I(hkl)>|/\Sigma_{hkl} \Sigma_j I(hkl)_j$, where $I(hkl)_j$ is the j[th] measurement of the intensity of reflection hkl and $<I(hkl)>$ is the average intensity.
[e]$R = \Sigma_{hkl}||F_{obs}| - |F_{calc}||/\Sigma_{hkl} |F_{obs}|$, where $R_{free}$ is calculated without a sigma cutoff for a randomly chosen 5% of reflections, which were not used for structure refinement and $R_{work}$ is calculated for the remaining reflections.
[f]Deviations from ideal bond lengths/angles.
[g]Percentage of residues in favored region/allowed region/outlier region.

concentration of 5 mM and annealed by mixing equal volumes of each strand to yield a final DNA duplex concentration of 2.5 mM. The DNA was heated to 95°C for 5 min and allowed to cool slowly to room temperature. Purified CBCs were added to a 1.2-fold molar excess of the respective DNA duplex. CBC (wt)–DNA mixtures were subjected to crystallization without additional purification steps. CBC[P88L]–DNA preparations were further purified by SEC in 20 mM Tris/HCl, 150 mM NaCl, 1 mM DTT, pH 7.5, using a Superdex prep grade 75 16/60 column (GE Healthcare). The presence of all three CBC subunits and DNA in the collected main fraction was verified by a dual stain method that allows independent visualization of the protein and nucleic acid species (Pryor et al, 2012). In brief, SDS–PAGE gels were first washed with water

followed by staining for protein with GelCode Blue Stain Reagent (Thermo Fisher Scientific). Next, the gels were again washed with water, followed by staining with 1× SYBR Gold Nucleic Acid Gel Stain (Invitrogen). SEC-purified CBC[P88L]–DNA preparations were concentrated 10-fold by ultrafiltration (Amicon Ultra-15 30K centrifugal filter device; Millipore) and subjected to crystallization.

### Crystallization and structure determination

All complexes were crystallized by the sitting drop vapor diffusion technique at 20°C. Crystal drops (0.4 µl) contained equal volumes of

the protein–DNA complex (13–15 mg ml$^{-1}$) and reservoir solution. All DNA fragments that encoded promoter sequences of either cyto-chrome c (*cycA*) or the vacuolar iron transporter (*cccA*) were 23-bp long and carried 5′ AA-TT overhangs.

Crystals of the An_CBC–*cccA* complex grew from 0.2 M ammonium acetate, 0.1 M 4-(2-hydroxyethyl)-1-piperazineethanesulfonic acid (Hepes), pH 7.5, and 25% (wt/vol) polyethylene glycole (PEG) 3350. The Afu_CBC–*cycA* structure was obtained from 0.1 M MES, pH 6.5, and 25% (wt/vol) PEG8000 and the Afu_CBC–*cccA* complex crys-tallized from 0.1 M MES, pH 6.5, and 25% (wt/vol) PEG6000. Crystals of the An_CBC$^{P88L}$–*cycA* complex grew from conditions containing 175 mM di-ammonium phosphate and 19% (wt/vol) PEG3350. All crystals were cryoprotected by the addition of a 1:1 (vol/vol) mixture of mother liquor and 70% (vol/vol) glycerol and subsequently super-cooled in a stream of nitrogen gas at 100 K. Diffraction data were collected at the beamline X06SA, Swiss Light Source at $\lambda$ = 1.0 Å. Reflection intensities were analyzed with the program package XDS (Kabsch, 1993). Structure determination was performed by Patter-son search calculations with PHASER (McCoy et al, 2007) using the coordinates of either wt *A. nidulans* CBC without DNA (PDB ID 4G91 [Huber et al, 2012]) or bound DNA (PDB ID 4G92 [Huber et al, 2012]). Cyclic refinement and model building steps were performed with REFMAC5 (Vagin et al, 2004) and Coot (Emsley et al, 2010). Water molecules were placed with ARP/wARP solvent (Perrakis et al, 1997). Translation/libration/screw refinements finally yielded good values for R$_{crys}$ and R$_{free}$ as well as rmsd bond and angle values. The models were proven to fulfill the Ramachandran plot using PROCHECK (Laskowski et al, 1993) (Table 1). The DNA-bending angle was analyzed with the Curves+ and SUMR algorithms (Lavery et al, 2009). Graphical illustrations were created with the UCSF Chimera package from the Resource for Biocomputing, Visualization, and Informatics at the University of California, San Francisco (Pettersen et al, 2004). Coor-dinates and structure factors have been deposited in the Protein Data Bank (for entry codes see Table 1).

## Data Availability

Coordinates and structure factors have been deposited in the Protein Data Bank under the entry codes 6Y35 (Afu_CBC–*cycA*), 6Y36 (Afu_CBC–*cccA*), 6Y37 (An_CBC–*cccA*), and 6Y39 (An_CBC HapE$^{P88L}$–*cycA*).

## Supplementary Information

## Acknowledgements

This work was funded by the joint D-A-CH program "Novel molecular mechanisms of iron sensing and homeostasis in filamentous fungi" (Deutsche Forschungsgemeinschaft [DFG] to AA Brakhage [BR 1130/14-1], M Groll [GR 1861/8-1], and P Hortschansky [HO 2596/1-1] and Austrian Science Foundation [FWF-I1346] to H Haas). MJ Bromley received funding from the Wellcome trust grant 208396/Z/17/Z and the National Institutes of Health (R01AI143198-01). EM Huber acknowledges funding by the DFG–SFB 1309–325871075 and the Young Scholar's Program of the Bavarian Academy of Sciences and Humanities. We thank the staff of the beamline X06SA at the Paul Scherrer Institute, Swiss Light Source, Villigen Switzerland for assis-tance during data collection, and acknowledge funding from the European Community's Seventh Framework Programme (FP7/2007-2013) under BioStruct-X (grant agreement no. 283570).

## Author Contributions

P Hortschansky: funding acquisition, investigation, visualization, project administration, and writing—review and editing.
M Misslinger: investigation and writing—review and editing.
J Mörl: investigation.
F Gsaller: investigation and writing—review and editing.
MJ Bromley: resources, supervision, funding acquisition, and wri-ting—review and editing.
AA Brakhage: conceptualization, funding acquisition, and writing—review and editing.
M Groll: conceptualization, funding acquisition, investigation, and writing—review and editing.
H Haas: conceptualization, supervision, funding acquisition, project administration, and writing—review and editing.
EM Huber: funding acquisition, investigation, visualization, project administration, and writing—original draft, review, and editing.

## Conflict of Interest Statement

The authors declare that they have no conflict of interest.

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
