## [Reviewer comments · Life Science Alliance]

Life Science Alliance

Structural basis of HapE-P88L-linked antifungal triazole resistance in *Aspergillus fumigatus*

Peter Hortschansky, Matthias Misslinger, Jasmin Mörl, Fabio Gsaller, Michael Bromley, Axel Brakhage, Michael Groll, Hubertus Haas, and Eva Huber

DOI: <https://doi.org/10.26508/lsa.202000729>

Corresponding author(s): Eva Huber, Technical University Munich

Review Timeline:

Submission Date:	2020-04-02
Editorial Decision:	2020-04-27
Revision Received:	2020-05-08
Accepted:	2020-05-12

Scientific Editor: Andrea Leibfried

Transaction Report:

April 27, 2020

RE: Life Science Alliance Manuscript #LSA-2020-00729-T

Dr. Eva M Huber
Technical University Munich
Department of Chemistry
Lichtenbergstraße 4
Garching 85748
Germany

Dear Dr. Huber,

Thank you for submitting your manuscript entitled "Structural basis of HapE-P88L-linked antifungal triazole resistance in *Aspergillus fumigatus*" to Life Science Alliance. Your work was assessed by expert reviewers now, and you can find their reports below.

As you will see, the reviewers appreciate your analyses and only raise very few concerns. We would thus be happy to publish your work in Life Science Alliance, pending minor revision:

- please address the reviewer comments by text changes; all three reviewer agreed during cross-commenting that testing an P88L allele in *A. nidulans* is not mandatory at this stage
- please fill in all mandatory fields in our submission system
- please link your ORCID iD to your profile in our submission system, you should have received an email with instructions on how to do so
- please upload all figures, including supplementary figures, as individual files; all figure legends should be only provided within the main manuscript text (incl. suppl. Figure legends)
- please add panel descriptors to the legend of figure 2
- please make sure that Table 1 remains in the main manuscript word docx file; if you would like to rather upload the table separately, please do so in excel or word docx format
- we display suppl. Figures in-line in the HTML version of the paper - there is thus no need to list Huber et al. as a suppl reference

A. FINAL FILES:

B. MANUSCRIPT ORGANIZATION AND FORMATTING:

Thank you for your attention to these final processing requirements.

Sincerely,

Andrea Leibfried, PhD
Executive Editor

Life Science Alliance
Meyerhofstr. 1
69117 Heidelberg, Germany
t +49 6221 8891 502
e a.leibfried@life-science-alliance.org
www.life-science-alliance.org

Reviewer #1 (Comments to the Authors (Required)):

In the present manuscript, Hortschansky et al. investigated the molecular mechanisms by which a recently discovered P88L exchange in the HapE subunit of the CCAAT-binding complex (CBC) in *Aspergillus* species leads to resistance against azoles. Such resistances constitute a growing health concern in clinics.

The authors conducted growth assays to assess consequences of the HapE(P88L) variant, showing that HapE(P88L) leads to growth defects in the presence of low or high iron levels, and showing that these growth defects correlate with reduced CBC-mediated expression of siderophore transporter MirB, lower levels of certain siderophores and dysregulation of iron-dependent proteins. HapE(P88L) also led to abrogated iron-dependent voriconazole resistance of spores. Extensive SPR analyses of DNA binding by wt and mutant CBCs showed decreased affinity for CBC containing HapE(P88L). The authors clarified the structural basis of the effects by determining crystal structures of several wt and HapE(P88L)-CBC/DNA complexes, observing changes in binding stoichiometry, mode of DNA association and DNA bending. Strikingly, they could trace the origin of altered DNA binding and bending to the altered residue leading to an extension of a helix that would clash with bent DNA. As a consequence, a normally DNA-binding element in another subunit lacks DNA contacts and remains disordered.

The work conducted appears to be technically sound, the results presented are of excellent quality and support the conclusions drawn. The manuscript is very well and effectively written and should be accessible to a broad audience. The work presented clarifies the molecular basis of an antifungal resistance mutation at the atomic level and is thus of biomedical relevance.

This is an elegant study presenting clear-cut results and this reviewer has no suggestions for improvements or points of criticism.

Reviewer #2 (Comments to the Authors (Required)):

Azole resistance in *Aspergillus fumigatus* (Afu) was thought to be caused by mutations within the gene (*cyp51A*) encoding the target enzyme for this drug class. As these authors point out, at least 50% if not more, of azole resistant Afu isolates have no linked defect in the coding sequence of *cyp51A* gene. One of the first mutations identified that led to azole resistance in a strain containing a wild-type *cyp51A* gene was a lesion within the *hapE* gene. This *hapE* allele will be referred to P88L for the associated amino acid change. The HapE protein is one of the subunits of the Afu CCAAT-binding complex (CBC) that may regulate 30% of all promoters in the cell. Interestingly, the CBC can act as a repressor or activator depending on promoter context. In Afu, the role of the CBC in azole resistance seems primarily to be via its action as a repressor of the transcription of *cyp51A*. This manuscript addresses the molecular mechanism underlying the increased azole resistance supported by the P88L allele of *hapE*.

Convincing data are presented that argue this hapE allele can still bind to its target sites in the genome but this binding is disturbed. The P88L form of HapE is still functional as the phenotypes supported by this mutant protein are not equivalent to a hapC null mutant lacking the CBC. Loss of hapX antagonizes the P88L phenotypes, providing an in vivo correlate to the importance of the interaction between the CBC and HapX. Additionally, in vitro DNA binding measurements demonstrate that the presence of recombinant HapX can partially suppress the observed DNA binding defect caused by the P88L HapE. Finally, crystallographic data are provided that indicate the P88L form of HapE has an altered 3 dimensional structure that may explain the altered properties of this mutant CBC.

I found this manuscript to be well-written and interesting to read. The work is excellent and provides new insight into the understanding of the behavior of the interesting P88L hapE allele. The authors argue that formation of alleles of the CBC that can elevate drug resistance is restricted due to the requirement that many mutants might cause too severe of a global defect due to the wide range of CBC target genes is compelling. This could explain why CBC nulls are not found and P88L is the only azole resistant form of the CBC known.

I think this manuscript could be accepted as is. My only suggestion for the authors is to consider constructing a P88L allele in *A. nidulans* to ensure that the phenotype is the same as in *A. fumigatus*. Given the 86% identity (!-I checked), this seems very likely to be true. Since the P88L allele was isolated in *A. fumigatus*, yet much of the work reported here is done with the *A. nidulans* factor, it would be nice to ensure that the phenotypes are also observed in *A. nidulans*. However, this is only a suggestion and should be left to the discretion of the authors as a large amount of work is already reported here.

Reviewer #3 (Comments to the Authors (Required)):

Emerging of drug-resistant *Aspergillus* pose a serious threat to food security and human health. In the light of recent events, secondary lung infections are of most concern.

While emerging mutations in the drug target gene are the main line of drug resistance, de-repression of the transcription regulatory pathways often leads to multi-drug resistance through over-expression of the drug target and efflux pumps.

The aim of this study was to evaluate the role of P88L mutation in HapE subunit of CCAAT motif - binding complex (CBC) in *Aspergillus*. This transcription factor controls the range of key metabolic genes. If deregulated, fitness, virulence and drug resistance of the pathogen could be altered. The main gene of interest *Cyp51a* is responsible for pan-azole resistance. In addition, the role of subunit HapX involved in iron sensing and in stabilizing the CBC was investigated.

Mechanistic basis of the effect of HapEP88L in azole resistance of *Aspergillus* spp was investigated in three groups of experiments.

In vivo: Isogenic wild type stains, mutants and knockouts of *A. fumigatus* were tested for iron requirement and tolerance, fitness, gene expression, siderophore production and azole resistance.

In vitro: Affinity of purified CBC complexes alone (for *A. fumigatus* and *A. nidulans*) and in combination with HapX (for *A. fumigatus*) to model DNA response elements evaluated by surface plasmon resonance.

Finally, CBC recombinant core domains from *A. nidulans* were co-crystallized with DNA targets with their structure examined.

Authors conclude that HapEP88L mutation weakens the binding of the repressor complex, and, therefore, de-represses among the others, the *cyp51A* gene. HapX was found to assist the repressor binding, which explains decreased azole resistance in the iron-deficient environment. While HapX subunit partially stabilizes the mutant form of the regulatory complex, this effect is weak, and therefore makes effect of iron content in the media irrelevant.

In general, the article is well written, structured and illustrated. Span, novelty and quality of the references provide adequate support for the story. The data provide good support for the conclusions. The discussion is clear and comprehensive.

Although, a few issues need to be addressed prior to publication.

Main issues:

The last statement of the abstract "Altogether, these results suggest that resistance [of the CBC mutant] to azoles will be enhanced in iron-deficient clinical niches such as the lung" is not clear enough. It can be interpreted in the way that low concentrations of iron would enhance resistance of the mutant. Although SBS mutant is more resistant to azoles in the -Fe conditions in comparison to the wild type, this resistance is iron-independent, and associated with ~ 50% less siderophore production (iron in the tissue is not entirely in the free form as in the experiment).

Figure 1C shows substantial decrease of Cyp51A expression for HapEP88L in the presence of 0.03mM Fe in comparison to iron deficient (-Fe) variant. This contradicts with the azole resistance pattern (Fig1E), and general conclusions and needs to be interpreted.

SPR results suggest that binding of mutated CBC to *cccA* and *cycA* decreased drastically for both *A. fumigatus* (Afu/AfuccA) and *A. nidulans*: An/AncycA (Page 6, Figure 2). In the SEC experiments (Page 7, Figure 4) poor association was confirmed for mutant Afu/AfuccA but not for An/AncycA where SEC peak #2 was tall. What will be the interpretation? Abnormality of 4A pattern is intriguing. Was the variant AfuP88L/AccccA (Fig2 2C) ever assessed by SEC and, if so, was it similar to 4A?

DNA band from fraction #1 (agarose gel, Fig 4A) runs above the others. Is this an artefact?

Technical issues:

Page 6 Line 5: (and Figure 1 legend) "... resistance of *A. fumigatus* spores:" although the spores were seeded on the plate, not spores but actively growing mycelia were the subject of resistance.

Page 23 Lines14 and 23: (Preparation of CBC:DNA complexes) "CBCP88L:DNA preparations were further purified" and "SEC purified CBCP88L:DNA preparations were concentrated by ultrafiltration" - only mutant complexes are mentioned. What about wt ones? Were these preparations made before?

Methods: it might be logical to put "Surface plasmon resonance (SPR) measurements" method before "Preparation of CBC:DNA complexes for crystallization"

Figure 4: This figure is not annotated enough in the legend or in the main text. How about marking the peak/fraction of interest (#1 for 4A,B; #2 for 4C)? What is it in the lines "2(30 kDa/R)" of the SDS gel (4B, 4C)?

Is it possible to position the gel annotations in figure 4 A in the same way/orientation as for C and D?

Reviewer 1

Reviewer 1 had “no suggestions for improvements or points of criticism”.

Reviewer 2

According to reviewer 2 “this manuscript could be accepted as is.”

1. The only “suggestion for the authors is to consider constructing a P88L allele in *A. nidulans* to ensure that the phenotype is the same as in *A. fumigatus*...However, this is only a suggestion and should be left to the discretion of the authors as a large amount of work is already reported here.”

Because *A. nidulans* is a non-pathogenic mold, transcriptional control of its *cyp51A* gene has barely been studied so far. Sequence analyses of the *A. nidulans cyp51A* promoter however suggest that the binding site for the inducer SREBP is present, while the bipartite recognition sequence for the CBC and HapX is missing (Furukawa *et al*, 2020). Thus, in *A. nidulans*, the function of the CBC in the context of *cyp51A* gene expression as well as azole resistance is unclear. Due to this lack of functional understanding in *A. nidulans*, construction of a P88L mutant *A. nidulans* strain would not provide any valuable conclusions for the present study and likely create more questions than answers. Furthermore, according to the editor “all three reviewers agreed during cross-commenting that testing an P88L allele in *A. nidulans* is not mandatory at this stage”.

Reviewer 3

Reviewer 3 states that “the article is well written, structured and illustrated. Span, novelty and quality of the references provide adequate support for the story. The data provide good support for the conclusions. The discussion is clear and comprehensive. Although, a few issues need to be addressed prior to publication”.

1. The last statement of the abstract “Altogether, these results suggest that resistance [of the CBC mutant] to azoles will be enhanced in iron-deficient clinical niches such as the lung” is not clear enough. It can be interpreted in the way that low concentrations of iron would enhance resistance of the mutant. Although SBS mutant is more resistant to azoles in the -Fe conditions in comparison to the wild type, this resistance is iron-independent, and associated with ~ 50% less siderophore production (iron in the tissue is not entirely in the free form as in the experiment).

We are grateful for this comment and changed the last sentence of the abstract to “Altogether, these results indicate that the mutation HapE^{P88L} confers increased resistance to azoles compared to wt *A. fumigatus*, particularly in low-iron clinical niches such as the lung.”

2. Figure 1C shows substantial decrease of Cyp51A expression for HapEP88L in the presence of 0.03mM Fe in comparison to iron deficient (-Fe) variant. This contradicts with the azole resistance pattern (Fig1E), and general conclusions and needs to be interpreted.

We agree with this referee, but there are several possible explanations for this contradictory effect: 1) The mRNA levels do not essentially correlate with the protein levels present in the cell. 2) Activity of Cyp51A, and consequently azole resistance, are dependent on heme bound iron as a cofactor. Hence, high *cyp51A* mRNA levels during iron starvation are not necessarily associated with high enzyme activity/increased azole resistance. Especially if iron as a cofactor is limiting and cofactor occupancy at the Cyp51A active site may be low. 3) For stable activity of Cyp51A, continuous supply with electrons delivered by the cytochrome b5 CybE and NAD(P)H is required. Expression of *cybE* however is CBC:HapX regulated and may also be affected by the HapE^{P88L} mutation, thus allowing for only little enzymatic activity despite high mRNA levels (Misslinger *et al*, 2017). 4) Apart from Cyp51A, ergosterol biosynthesis involves other iron-dependent CBC-regulated enzymes (Gsaller *et al*, 2016). Consequently, under iron starvation, ergosterol biosynthesis and azole resistance are lower than under iron sufficiency conditions – irrespective of high *cyp51A* mRNA levels. 5) *A. fumigatus* also encodes a paralog of Cyp51A, namely Cyp51B, whose activity might contribute to azole resistance as well (Buied *et al*, 2013; Hagiwara *et al*, 2016).

3. SPR results suggest that binding of mutated CBC to *cccA* and *cycA* decreased drastically for both *A. fumigatus* (Afu/AfuccA) and *A. nidulans*: An/AncycA (Page 6, Figure 2). In the SEC experiments (Page 7, Figure 4) poor association was confirmed for mutant Afu/AfuccA but not for An/AncycA where SEC peak #2 was tall. What will be the interpretation? Abnormality of 4A pattern is intriguing. Was the variant AfuP88L/AccccA (Fig2 2C) ever assessed by SEC and, if so, was it similar to 4A?

We noted reduced affinities of AfuCBCHapE^{P88L} and AnCBCHapE^{P88L} protein complexes for An_*cycA*, Afu_*cccA* and An_*cccA* promoter derived sequences in our SPR analyses. However, the affinities differed and had the following tendency: An_*cccA* (~2 μM) << Afu_*cccA* (~400 nM) < An_*cycA* (~100 nM; Figure 2). Because the An_*cccA* promoter sequence showed poor affinity for both Afu and An mutant CBC, we did not consider these samples further and thus cannot provide any SEC analyses. Samples with the Afu_*cccA* DNA, which showed intermediate affinity in the SPR analyses, did not yield any stable protein:DNA complexes according to SEC (Figure 4A). Differences between SPR and SEC experiments probably result from the shearing forces that act on the protein:DNA complex during chromatography. Although weak association is possible in solution, shearing forces might have disrupted the complexes during SEC. For An_*cycA* DNA fragments, we noted the highest affinity (~100 nM, Figure 2) and the resulting complexes were also stable to SEC (Figure 4B, C). Thus, the affinity of protein to DNA was obviously higher than the shearing forces. We have amended the text describing the SEC experiments accordingly and changed the order of the panels in Figure 2 to enhance perceivability in conjunction with its legend:

Azole resistant Afu_CBC^{P88L} protein preparations however did not stably associate with Afu_cccA promoter-derived double stranded DNA, as confirmed by size exclusion chromatography (Figure 4A). The residual complex affinity of 429 nM (Figure 2C) probably was not high enough to counteract the shearing forces during chromatography. In addition, Afu_CBC^{P88L} failed to crystallize in the presence of An_cycA promoter DNA (Figure 4B). We therefore switched organisms and created the HapE^{P88L} mutant *A. nidulans* CBC. Despite reduced affinity (116 nM versus 0.83 nM for wt; Figure 2A), we obtained a SEC-stable complex for this variant with the 23 bps long *cycA* promoter fragment (Figure 4C), and elucidated its X-ray structure at 2.3 Å resolution (Table 1).

4. DNA band from fraction #1 (agarose gel, Fig 4A) runs above the others. Is this an artefact?

Figure 4 shows SDS-PAGEs of the respective protein:DNA samples. We do not know why the DNA band from fraction #1 in Figure 4A is slightly shifted. Maybe such artefacts can arise when DNA fragments are separated by SDS-PAGE. However, since the samples from Figure 4A have not been used for further studies, this effect is of no importance for the whole study.

5. Page 6 Line 5: (and Figure 1 legend) "... resistance of *A. fumigatus* spores:" although the spores were seeded on the plate, not spores but actively growing mycelia were the subject of resistance.

We are grateful for this comment and changed the text accordingly:

Page 6, line 5: "Next, we tested the resistance of *A. fumigatus* to the broad-spectrum antifungal medication voriconazole."

Figure 1, legend: "The narrower the inhibition zone was, the more resistant the strains were."

6. Page 23 Lines 14 and 23: (Preparation of CBC:DNA complexes) "CBCP88L:DNA preparations were further purified" and "SEC purified CBCP88L:DNA preparations were concentrated by ultrafiltration" - only mutant complexes are mentioned. What about wt ones? Were these preparations made before?

We now provide more details on the preparation of the CBC (wt):DNA complexes in the methods section (pages 19 and 20):

"Size exclusion fractions containing pure CBCs were pooled based on SDS-PAGE analysis, concentrated by ultrafiltration (Amicon Ultra-15 10K centrifugal filter device, Millipore) to 16-18 mg mL⁻¹, aliquoted, flash frozen in liquid nitrogen, and stored at -80 °C."...

"Purified CBCs were added to a 1.2-fold molar excess of the respective DNA duplex. CBC (wt):DNA mixtures were subjected to crystallization without additional purification steps. CBC^{P88L}:DNA preparations were further purified by size exclusion chromatography (SEC) in 20 mM Tris/HCl, 150 mM NaCl, 1 mM DTT, pH 7.5 using a Superdex prep grade 75 16/60 column (GE Healthcare)."...

"SEC purified CBC^{P88L}:DNA preparations were concentrated 10-fold by ultrafiltration (Amicon Ultra-15 30K centrifugal filter device, Millipore) and subjected to crystallization."

7. *Methods: it might be logical to put "Surface plasmon resonance (SPR) measurements" method before "Preparation of CBC:DNA complexes for crystallization"*

We changed the order of the methods according to the suggestion of reviewer 3.

8. *Figure 4: This figure is not annotated enough in the legend or in the main text. How about marking the peak/fraction of interest (#1 for 4A,B; #2 for 4C)? What is it in the lines "2(30 kDa/R)" of the SDS gel (4B, 4C)?*

We expanded the legend of Figure 4 by providing additional information. In particular, the fraction that was concentrated and used for crystallization is now appropriately labeled and colored in red.

9. *Is it possible to position the gel annotations in figure 4 A in the same way/orientation as for C and D?*

In all panels, the labels have now been positioned the same way.

References

- Buied A, Moore CB, Denning DW, Bowyer P (2013) High-level expression of cyp51B in azole-resistant clinical *Aspergillus fumigatus* isolates. *J Antimicrob Chemother* 68: 512-514
- Furukawa T, Scheven MT, Misslinger M, Zhao C, Hoefgen S, Gsaller F, Lau J, Jöchl C, Donaldson I, Valiante V *et al* (2020) The fungal CCAAT-binding complex and HapX display highly variable but evolutionary conserved synergetic promoter-specific DNA recognition. *Nucleic acids research*: 10.1093/nar/gkaa1109
- Gsaller F, Hortschansky P, Furukawa T, Carr PD, Rash B, Capilla J, Muller C, Bracher F, Bowyer P, Haas H *et al* (2016) Sterol Biosynthesis and Azole Tolerance Is Governed by the Opposing Actions of SrbA and the CCAAT Binding Complex. *PLoS Pathog* 12: e1005775
- Hagiwara D, Watanabe A, Kamei K, Goldman GH (2016) Epidemiological and Genomic Landscape of Azole Resistance Mechanisms in *Aspergillus* Fungi. *Front Microbiol* 7: 1382
- Misslinger M, Gsaller F, Hortschansky P, Muller C, Bracher F, Bromley MJ, Haas H (2017) The cytochrome b5 CybE is regulated by iron availability and is crucial for azole resistance in *A. fumigatus*. *Metallomics* 9: 1655-1665

May 12, 2020

RE: Life Science Alliance Manuscript #LSA-2020-00729-TR

Dr. Eva M Huber
Technical University Munich
Department of Chemistry
Lichtenbergstraße 4
Garching 85748
Germany

Dear Dr. Huber,

Thank you for submitting your Research Article entitled "Structural basis of HapE-P88L-linked antifungal triazole resistance in *Aspergillus fumigatus*". I appreciate the introduced changes and it is a pleasure to let you know that your manuscript is now accepted for publication in Life Science Alliance. Congratulations on this interesting work.

DISTRIBUTION OF MATERIALS:

Again, congratulations on a very nice paper. I hope you found the review process to be constructive and are pleased with how the manuscript was handled editorially. We look forward to future exciting submissions from your lab.

Sincerely,

Andrea Leibfried, PhD
Executive Editor
Life Science Alliance
Meyerohofstr. 1
69117 Heidelberg, Germany
t +49 6221 8891 502
e a.leibfried@life-science-alliance.org
www.life-science-alliance.org